# LIX1 Controls MAPK Signaling Reactivation and Contributes to GIST-T1 Cell Resistance to Imatinib

**DOI:** 10.3390/ijms24087138

**Published:** 2023-04-12

**Authors:** Salomé Ruiz-Demoulin, Eva Trenquier, Sanaa Dekkar, Sébastien Deshayes, Prisca Boisguérin, César Serrano, Pascal de Santa Barbara, Sandrine Faure

**Affiliations:** 1Physiology and Experimental Medicine of the Heart and Muscles (PhyMedExp), University of Montpellier, INSERM, CNRS, 34295 Montpellier, France; 2Sarcoma Translational Research Laboratory, Vall d’Hebron Institute of Oncology (VHIO), 08035 Barcelona, Spain

**Keywords:** LIX1 resistance, tyrosine kinase inhibitors, gastrointestinal tumors, MAPK

## Abstract

Gastrointestinal stromal tumor (GIST), the most common sarcoma, is mainly caused by an oncogenic mutation in the KIT receptor tyrosine kinase. Targeting KIT using tyrosine kinase inhibitors, such as imatinib and sunitinib, provides substantial benefit; however, in most patients, the disease will eventually progress due to KIT secondary mutations leading to treatment failure. Understanding how GIST cells initially adapt to KIT inhibition should guide the selection of appropriate therapies to overcome the emergence of resistance. Several mechanisms have been broadly implicated in the resistance to imatinib anti-tumoral effects, including the reactivation of MAPK signaling upon KIT/PDGFRA targeted inhibition. This study provides evidence that LImb eXpression 1 (LIX1), a protein we identified as a regulator of the Hippo transducers YAP1 and TAZ, is upregulated upon imatinib or sunitinib treatment. LIX1 silencing in GIST-T1 cells impaired imatinib-induced MAPK signaling reactivation and enhanced imatinib anti-tumor effect. Our findings identified LIX1 as a key regulator of the early adaptative response of GIST cells to targeted therapies.

## 1. Introduction

A gastrointestinal stromal tumor (GIST), the most common mesenchymal neoplasm of the gastrointestinal tract, originates from interstitial cells of Cajal (ICC) or related mesenchymal progenitors that require an elevated KIT expression for lineage specification and survival [1,2,3,4]. A GIST occurs predominantly in the stomach (50–60%) and small intestine (30–35%) and is mainly driven by activating mutations (present in 85–90% of patients) in the receptor tyrosine kinases KIT and platelet-derived growth factor receptor A (PDGFRA). Oncogenic KIT mutations are found in approximately 80% of sporadic GISTs [5], but familial syndromes harboring germline-activating KIT mutations have been described. These patients develop diffuse ICC hyperplasia that eventually progresses to GIST [6]. The KIT proto-oncogene encodes a class III receptor-type tyrosine kinase that is activated upon binding to its cognate ligand, stem cell factor, via its extracellular domain [7]. This leads to receptor homo-dimerization and activation of the intracellular kinase domain that consequently initializes downstream signaling, such as the PI3K–AKT–mTOR and RAS–MAPK pathways. These pathways are implicated in regulating cellular functions, especially in ICCs where KIT physiologic activity is indispensable for cell proliferation, differentiation, and apoptosis [8]. Primary KIT mutations mainly occur in exon 11 (70–80%) that encodes the juxtamembrane domain. This leads to disruption of the auto-inhibitory function, resulting in constitutive, ligand-independent kinase KIT activity and constitutive activation of downstream KIT-activated AKT and MAPK signaling. Both pathways are crucial for GIST initiation and tumor development by exerting a critical regulation of cancer cell proliferation and apoptosis evasion [9,10,11].

Tyrosine kinase inhibitors (TKI), such as imatinib, inhibit KIT-downstream PI3K and MAPK signaling and consequently impair the viability of GIST cells in which KIT signaling is constitutively activated [12]. Patients with unresectable or metastatic GIST show very good clinical responses to TKI [13]. However, despite the early clinical success, complete response in patients treated with first-line imatinib is rarely achieved, and prolonged treatment is required to avoid disease progression. This often leads to the appearance of secondary resistance mutations in KIT after approximately 18–24 months of treatment [9,14]. Most frequently, disease progression is explained by the emergence of polyclonal subpopulations with secondary KIT kinase-domain mutations that decrease imatinib binding affinity [15,16,17]. Therefore, other TKIs, such as sunitinib and regorafenib, are used as a second- and third-line treatment, respectively, after imatinib failure [18,19]. However, the clinical benefit achieved by these treatments is modest, with progression-free survival of <6 months and response rates <10% [18,19,20,21,22]. Several mechanisms have been implicated in the adaptative response of GIST cells to targeted therapies, including the reactivation of pathways downstream of KIT. For instance, reactivation of MAPK signaling, through activation of fibroblast growth factor receptor (FGFR) 1 and 2 signaling or through receptor tyrosine–kinase switch, decreases the imatinib anti-tumor effect [23,24,25]. Given the importance of MAPK signaling in the early adaptation of GIST cells to imatinib, a potential treatment strategy would be the combined inhibition of KIT and MAPK pathways to prevent the emergence of imatinib-resistant clones in patients with GIST [26,27].

In this study, we focused on LImb eXpression 1 (LIX1), a protein that we previously identified as a regulator of digestive mesenchymal progenitor proliferation upstream of the Hippo transducers YAP1 and TAZ [28]. In GIST, LIX1 controls mitochondrial function, KIT protein level, ICC lineage specification through YAP1/TAZ, and cell proliferation [29,30]. Importantly, LIX1 expression is higher in patients with relapsed GIST [23]. This finding prompted us to examine LIX1 contribution to TKI resistance in GIST. We found that the LIX1 level was upregulated in GIST cell lines upon incubation with imatinib or sunitinib. Mechanistically, we found that LIX1 promoted a rebound of MAPK activation upon KIT/PDGFRA-targeted inhibition that leads to a reduction of the TKI effect. We then inhibited KIT (with imatinib) and/or LIX1 (by silencing) in an imatinib-sensitive GIST cell line and found that their combined inhibition further impaired cancer cell viability compared with cells incubated with imatinib alone. Thus, our study identified LIX1 as a new therapeutic target to prevent MAPK reactivation and overcome therapeutic adaptation in GIST.

## 2. Results

### 2.1. KIT-Signaling Abrogation Results in a Significant Increase in LIX1 Expression in GIST-T1 Cells

LIX1 is normally expressed in digestive mesenchymal progenitors only during fetal life, but its expression is high in GIST samples. When we analyzed the recurrence-free survival in the function of LIX1 expression, we found the highest LIX1 expression in relapsed tumors [29]. This prompted us to investigate LIX1’s contribution to the mechanisms of drug resistance. We first analyzed LIX1 expression in imatinib/sunitinib-sensitive (GIST-T1) and imatinib-resistant (GIST-T1/670) GIST cell lines upon blockade of KIT signaling using TKIs. The GIST-T1/670 cell line is a GIST-T1 clone isolated after continuous culture in 5 μM imatinib and acquired a secondary KIT kinase domain missense mutation (T670I in exon 14, leading to resistance to imatinib) in addition to the primary KIT exon 11 deletion [31,32]. GIST-T1/670 cells are resistant to imatinib but sensitive to sunitinib [33,34]. In GIST-T1 cells, the LIX1 mRNA level was significantly increased after 48 h exposure to imatinib and sunitinib (Figure 1A,B). The LIX1 protein level was also significantly increased upon inhibition of KIT signaling with imatinib (confirmed by the downregulation of phosphorylated KIT) (Figure 1C,D). Moreover, the baseline LIX1 expression (mRNA and protein) was higher in GIST-T1/670 cells than in GIST-T1 cells (Figure 1E–G), and its expression further increased when we cultured GIST-T1/670 cells in the presence of sunitinib for 48 h (Figure 1H). Thus, LIX1 expression increases in GIST-T1 cells upon KIT-signaling blockade using imatinib and sunitinib (first- and second-line TKI for GIST, respectively).

### 2.2. LIX1 Expression and MAPK Signaling Changes Following Incubation with Imatinib

Understanding how GIST responds to imatinib at the beginning of treatment should guide the selection of appropriate strategies to overcome the later emergence of secondary KIT mutations. Therefore, we determined to what extent LIX1 was involved in the early adaptative response to imatinib. Indeed, tumors adapt to targeted therapies in a relatively short period of time [23]. To this aim, we monitored LIX1 expression in GIST-T1 cells incubated or not with imatinib at 4, 24, and 48 h of incubation. We observed a significant increase in LIX1 (mRNA and protein) expression at 24 h (Figure 2A,B,D). This was associated with a marked increase in YAP1/TAZ expression (Figure 2C,E). Several mechanisms have been broadly involved in the attenuation of imatinib anti-tumoral effect, including the reactivation of pathways downstream of KIT. Therefore, we evaluated the activity of the KIT, MAPK, and PI3K pathways by measuring the levels of total and phosphorylated KIT, ERK1/2, and AKT, respectively, by western blotting. Imatinib led to a decrease in KIT activity and consequently of the downstream MAPK and PI3K signaling at 4 h, as previously reported ([23]; Figure 2F–H). MAPK-signaling inhibition was further confirmed by the downregulation of the ERK-dependent genes *SPROUTY2* and *SPROUTY4* (Figure 2I,J). Unlike KIT-signaling inhibition, MAPK inhibition was not sustained in GIST-T1 cells incubated with imatinib. Indeed, ERK1/2 phosphorylation and activity (phosphorylated ERK/total ERK ratio) were significantly increased at 48 h of incubation (Figure 2F,H). MAPK-signaling reactivation was further confirmed by the significant increase in the expression of *SPROUTY2* and *SPROUTY4* at 48 h of incubation (Figure 2I,J). Thus, the LIX1 expression increase is an early event in GIST cell response to KIT inhibition.

### 2.3. LIX1 Promotes MAPK-Signaling Reactivation Following KIT Inhibition

To determine whether MAPK-signaling reactivation in imatinib-treated cells requires LIX1, we used GIST-T1 cell lines that stably express negative control shRNA (GIST-T1-*Scramble*) or two different shRNAs against LIX1 (GIST-T1-*ShLIX1#1* and GIST-T1-*ShLIX1#2*) [29]. We then quantified LIX1 expression at different time points by RT-qPCR analysis in GIST-T1-*Scramble* and GIST-T1-*ShLIX1* cells incubated or not with 0.5 μM imatinib. We confirmed LIX1 downregulation in GIST-T1-*ShLIX1* compared with GIST-T1-*Scramble* cells before the addition of imatinib (Figure 3A). Incubation with imatinib significantly increased LIX1 expression level starting at 24 h, in GIST-T1-*Scramble* and also in GIST-T1-*ShLIX1* cells (Figure 3A). Nevertheless, the LIX1 mRNA level remained low in treated GIST-T1-*ShLIX1* cells, at a level comparable to that of untreated GIST-T1-*Scramble* cells. In addition, we observed a marked increase of YAP1/TAZ expression in GIST-T1-*Scramble* cells from the 24-h time point but not in GIST-T1-*ShLIX1* cells (Figure 3B; Appendix A). Moreover, ERK1/2 activity (western blot analysis) increased again at 48 h in treated GIST-T1-*Scramble* cells but not in GIST-T1-*ShLIX1* cells (Figure 3B,C). MAPK inhibition was maintained in GIST-T1 cells in which LIX1 was silenced, as demonstrated by the similar expression levels of *SPROUTY2* and *SPROUTY4* at the 4 h and 48 h time points (Figure 3D,E). Thus, LIX1 is implicated in imatinib-induced MAPK-signaling reactivation in GIST-T1 cells.

### 2.4. LIX1 Blockade Re-Sensitizes GIST-T1 Cells to Imatinib

As MAPK-signaling reactivation upon KIT/PDGFRA inhibition decreases imatinib anti-tumoral effect [23], we asked whether silencing LIX1 in GIST-T1 cell lines exposed to imatinib may overcome this effect. Incubation with imatinib for 48 and 72 h decreased the viability of GIST-T1-*Scramble* cells and even more of GIST-T1-*ShLIX1* cells (Figure 4A–D). This indicated that the combination of LIX1 and KIT inhibition was more cytotoxic than KIT inhibition alone. We then exposed GIST-T1-*Scramble* and GIST-T1-*ShLIX1* cells to increasing concentrations of imatinib. Imatinib half-maximal inhibitory concentration (IC_50_) values were 54.95 nM in GIST-T1-*Scramble* cells and 26.81 nM and 15.74 nM in GIST-T1-*ShLIX1#1* and GIST-T1-*ShLIX1#2* cells (Figure 4E). Thus, LIX1 silencing potentiates the anti-tumor effect of imatinib.

## 3. Discussion

Although imatinib is a highly effective therapy against GIST, metastatic disease remains incurable. However, the majority of patients with GIST will eventually relapse. This, in turn, prompts the interest in understanding the biological mechanisms behind therapeutic adaptation to targeted inhibition of KIT in order to develop new treatment strategies for GIST cell eradication. GIST cells undergo cytostatic response to KIT inhibitors, which will eventually lead to the development of resistance in patients [31,33]. Imatinib induces GIST cell quiescence [31,33]. Its withdrawal leads to the cell cycle re-entry of residual quiescent cancer cells that start to proliferate, which is a major cause of disease progression. On the other hand, continuous imatinib treatment will lead to the emergence of polyclonal subpopulations harboring secondary KIT kinase-domain mutations that decrease imatinib binding affinity for this kinase [31,33]. Therefore, understanding how GIST cells adapt to KIT inhibition should allow the development of novel therapeutic strategies to overcome the appearance of secondary KIT mutations.

Imatinib treatment leads to KIT-signaling inhibition and consequently to MAPK downregulation ([23]; Figure 2F,H). Unlike KIT-signaling inhibition, MAPK inhibition is not sustained in GIST cells exposed to imatinib, and a rebound of ERK activity occurs shortly thereafter, thus hindering GIST cell eradication ([23]; Figure 2F,H). Feedback activation of FGF signaling could explain ERK rebound [23]. Indeed, the combination of BGJ398 (a pan-FGF receptor inhibitor) and imatinib represses ERK reactivation and enhances imatinib anti-tumor activity in GIST. However, this combination strategy exhibited high toxicity and limited its use in the clinic [35]. Therefore, the discovery of novel potential drivers remains an unmet clinical need.

In this study, we examined LIX1’s contribution to the therapeutic adaptation of GIST cells to imatinib. Human LIX1 is a highly conserved gene that encodes a 282-amino acid protein. In physiological conditions, LIX1 is expressed only during fetal life and controls the commitment of digestive mesenchymal progenitors and their plasticity [28]. Plasticity is often associated with higher cancer risk, as observed in GIST [3,4,36]. In GIST, LIX1 is overexpressed and is associated with poor prognosis. We previously demonstrated that LIX1 is a critical regulator of GIST development [29]. Here, we provide evidence that LIX1 promotes MAPK reactivation in GIST-T1 cells during treatment with imatinib. Accordingly, LIX1 silencing mimics the effects induced by MAPK inhibitors and enhances the imatinib anti-tumor effect. Thus, our work suggests that LIX1 could be a new therapeutic target to prevent MAPK reactivation and overcome TKI resistance in GIST. This research, however, is subject to the main limitation of having evaluated LIX1 only in GIST-T1 cells. It remains to understand how LIX1 controls MAPK reactivation in the presence of imatinib. LIX1 is localized in mitochondria, where it controls the shape of mitochondrial cristae and redox signaling [30]. It is well known that metabolic reprogramming is a hallmark of cancer cells to adapt to targeted therapy. Cells use two major metabolic pathways to produce the energy needed for their functions: (i) aerobic glycolysis, in which glucose is converted into pyruvate and lactate, and (ii) the mitochondrial oxidative phosphorylation (OXPHOS) machinery. This machinery is a key functional unit in mitochondria, and it combines electron transport with cell respiration and ATP synthesis. A consequence of mitochondrial oxidative metabolism is the generation of copious amounts of reactive oxygen species (ROS) by the electron transport chain. ROS activate signaling pathways that promote cancer cell proliferation and participate in genomic instability by inducing oxidative DNA damage leading to genomic mutations [37,38]. GIST cells display high glycolysis activity; however, upon incubation with imatinib, they shift to the OXPHOS machinery [39,40]. Interestingly, inhibition of mitochondria activity in the presence of imatinib forces a return to a glycolytic phenotype, a strategy that re-sensitizes GIST cells to imatinib [40]. Future experiments will determine whether LIX1 controls MAPK reactivation by modulating the metabolic phenotype of GIST cells in response to imatinib.

In conclusion, this work identified LIX1 as a key regulator of one of the early mechanisms leading to the adaptative response of GIST cells to target therapies. LIX1 inhibition could maximize the therapeutic response to imatinib treatment.

## 4. Materials and Methods

### 4.1. Cell Culture and Reagents

The GIST-T1 cell line, obtained from Cosmo Bio (Japan), was established from a metastatic human GIST that harbors a heterozygous deletion of 57 bases in exon 11 of KIT [41]. The imatinib-resistant GIST-T1/670 cell clone was derived from GIST-T1 cells upon culture in the presence of 5 μM imatinib and acquired a secondary missense T670I mutation in exon 14 of KIT [11,31,32]. GIST-T1 cell lines that stably express control shRNA (GIST-T1-*Scramble*) or shRNAs targeting two distinct regions of LIX1 (GIST-T1-*ShLIX1#1* and GIST-T1-*ShLIX1#2*) were previously developed [29]. Their characterization by RT-qPCR analysis has confirmed LIX1 downregulation in GIST-T1-*ShLIX1* cells with a higher efficacy of *ShLIX1#2* than *ShLIX1#1* [29]. All cells were cultured in Dulbecco’s Modified Eagle’s Medium (DMEM, Lonza, France) supplemented with 10% fetal bovine serum and 1% penicillin/streptomycin and routinely tested for the absence of mycoplasma contamination (Venor-GeM OneStep Test, BioValley). GIST cell lines were incubated with imatinib mesylate (STI571, Euromedex, France) and sunitinib (SU11248) malate (SE-S1042, Euromedex, France) at the concentrations indicated in the figure legends.

### 4.2. Immunoblot Analysis

Cell lysates were prepared as described previously (Guérin et al., 2020). Electrophoresis was carried out using 10 μg of extracts, loaded on 10% polyacrylamide gels, and transferred to nitrocellulose membranes. Primary antibodies used for immunoblot analysis are listed in Appendix A.

### 4.3. Reverse Transcription and Quantitative Polymerase Chain Reaction (RT-qPCR)

Total RNA extraction, reverse transcription, and qPCR analysis were performed as previously described [29]. PCR primers (listed in Appendix A) were designed using the LightCycler Probe Design 2.0 software. Expression levels were determined with the LightCycler analysis software (version 3.5) relative to standard curves. Data are the mean level of gene expression relative to the expression of the reference genes *HMBS* and *YWHAZ* calculated using the 2^−ΔΔCT^ method.

### 4.4. Cell Viability Measurement

Cell viability was analyzed 48 h and 72 h after TKI treatment by crystal violet staining (C6158, Sigma, France), as previously described [42]. Plated cells were washed with 1× PBS and incubated min with 0.1% crystal violet solution for 20 min. After several washes, 10% acetic acid solution was added to lyse-stained cells. Absorbance was measured at 595 nm using an Infinite 200 Pro Microplate Luminometer (Tecan Trading AG).

### 4.5. Statistical Analysis

Data were analyzed with the GraphPad Prism 9 software (GraphPad Software, Inc.). Statistical significance was calculated with the two-tailed Mann–Whitney test or, when appropriate, one-way ANOVA followed by Tukey’s multiple comparisons test (comparison between every mean value with all the other mean values) or Dunnett’s multiple comparisons test (comparison between every mean value with the control mean value), as indicated in the legends to figures. Results were considered significant when *p* < 0.05.

## Figures and Tables

**Figure 1 ijms-24-07138-f001:**
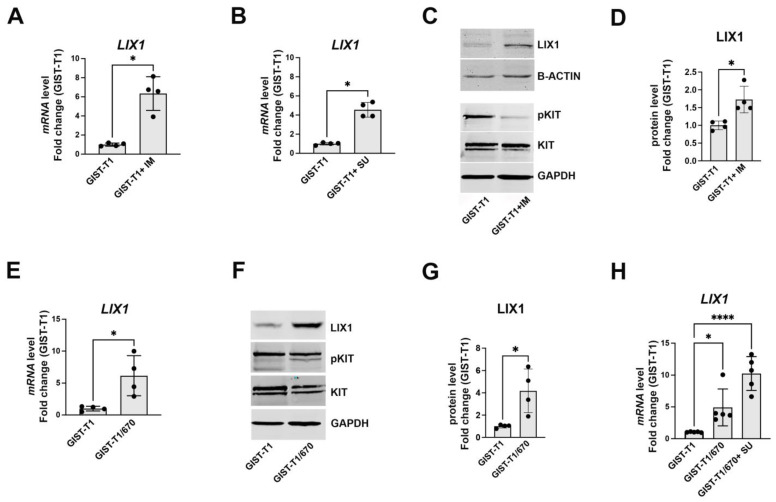
LIX1 expression is increased in GIST cells upon KIT-signaling inhibition using TKI. (**A**,**B**) RT-qPCR analysis of LIX1 transcript levels in GIST-T1 cells cultured in the presence of 0.5 μM imatinib (**A**) or 0.5 μM sunitinib for 48 h (**B**). Data were normalized to the mean *HMBS* and *YWHAZ* expression and converted to fold change. Values are the mean ± SEM of four independent experiments; * *p* < 0.05 (two-tailed Mann–Whitney test). (**C**) Representative western blot showing LIX1, phosphorylated KIT (pKIT), and KIT levels in GIST-T1 cells cultured with 0.5 μM imatinib for 48 h. Equal loading was verified by GAPDH expression. (**D**) Quantification of LIX1 level normalized to GAPDH level. Normalized expression levels were converted into fold change. Values are the mean ± SEM of four independent experiments; * *p* < 0.05 (two-tailed Mann–Whitney test). (**E**) RT-qPCR analysis of LIX1 transcript levels in GIST-T1 and imatinib-resistant GIST-T1/670 cells. Data were normalized to the mean *HMBS* and *YWHAZ* expression and converted to fold changes. Values are the mean ± SEM of four independent experiments; * *p* < 0.05 (two-tailed Mann–Whitney test). (**F**) Western blot analysis of endogenous LIX1 levels in GIST-T1 and GIST-T1/670 cells. (**G**) Quantification of LIX1 level normalized to GAPDH level. Normalized expression levels were converted into fold change. Values are the mean ± SEM of four independent experiments; * *p* < 0.05 (two-tailed Mann–Whitney test). (**H**) RT-qPCR analysis of LIX1 transcript levels in GIST-T1 cells, GIST-T1/670 cells, and GIST-T1/670 cells cultured in the presence of sunitinib for 48 h. Data were normalized to the mean *HBMS* and *YWHAZ* expression and converted to fold change. Values are the mean ± SEM of five independent experiments; * *p* < 0.05; **** *p* < 0.0001 (one-way ANOVA test followed by Dunnett’s multiple comparisons test).

**Figure 2 ijms-24-07138-f002:**
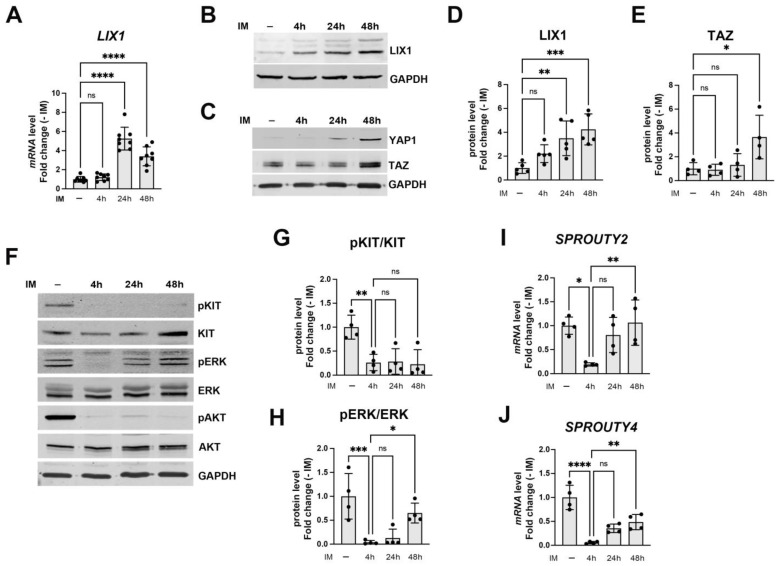
LIX1 expression at different time points after KIT inhibition with imatinib. GIST-T1 cells were cultured in the absence or presence of 0.5 μM imatinib (IM) and collected at 4, 24, and 48 h. Normalized expression levels were converted into fold change relative to control untreated cells (-). (**A**) RT-qPCR analysis of LIX1 transcript levels. Data were normalized to the mean *HMBS* and *YWHAZ* expression. Values are the mean ± SEM of eight independent experiments; **** *p* < 0.0001; ns, non-significant (one-way ANOVA followed by Dunnett’s multiple comparisons test). (**B**,**C**) Western blot analysis of endogenous LIX1 (**B**) and YAP1/TAZ protein levels (**C**). Equal loading was verified by GAPDH expression. (**D**,**E**) Quantification of LIX1 (**D**) and TAZ (**E**) protein levels normalized to GAPDH level. Values are the mean ± SEM of four independent experiments. * *p* < 0.05, ** *p* < 0.01, *** *p* < 0.001, **** *p* < 0.0001, ns, non-significant (one-way ANOVA followed by Dunnett’s multiple comparisons test). (**F**) Western blot analysis. Membranes were probed with antibodies against KIT, ERK, and AKT and their phosphorylated (p) forms, representative of KIT, MAPK, and PI3K pathway activities. Equal loading was verified by GAPDH expression. (**G**,**H**) Quantification of KIT (**G**) and MAPK (ERK) (**H**) signaling activity. Values were calculated as the phosphorylated/total protein ratio after normalization to the GAPDH level. For (**G**,**H**), values are the mean ± SEM of four independent experiments; * *p* < 0.05, ** *p* < 0.01, *** *p* < 0.001; ns, non-significant (one-way ANOVA followed by Tukey’s multiple comparisons test). (**I**,**J**) RT-qPCR analysis of *SPROUTY2* (**I**) and *SPROUTY4* (**J**) transcript levels. Data were normalized to the mean *HMBS* and *YWHAZ* expression. Values are the mean ± SEM of four independent experiments; * *p* < 0.05, ** *p* < 0.01, **** *p* < 0.0001; ns, non-significant (one-way ANOVA followed by Tukey’s multiple comparisons test).

**Figure 3 ijms-24-07138-f003:**
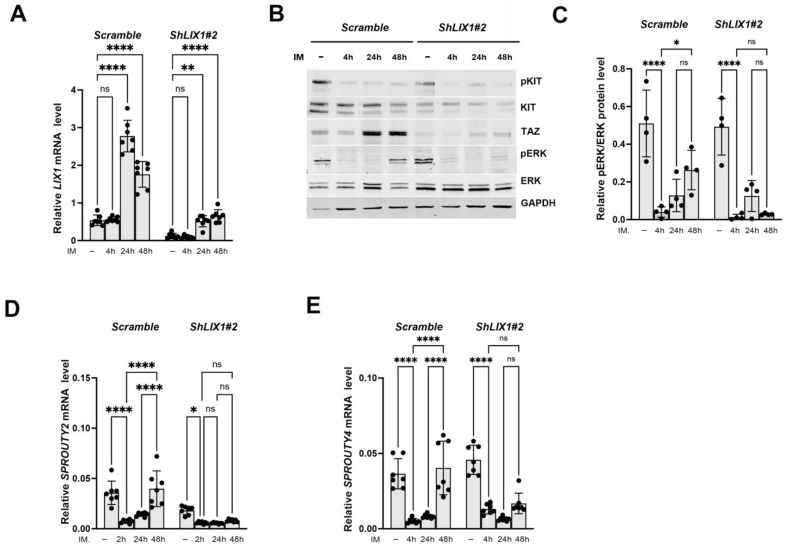
LIX1 promotes MAPK-signaling reactivation in imatinib-treated GIST cells. Control (GIST-T1-*Scramble*) or GIST-T1 cells in which LIX1 was silenced with shRNAs (GIST-T1-*ShLIX1#2*) were cultured in the absence (-) or the presence of 0.5 μM imatinib (IM) and collected after 4, 24, and 48 h. (**A**) RT-qPCR analysis of LIX1 transcript levels. Data were normalized to the mean *HMBS* and *YWHAZ* expression. Values are the mean ± SEM of seven independent experiments. ** *p* < 0.01, **** *p* < 0.0001; ns, non-significant (one-way ANOVA followed by Tukey’s multiple comparisons test). (**B**) Western blot analysis. Membranes were probed with antibodies against KIT, ERK, and AKT proteins and their phosphorylated forms. Equal loading was verified by GAPDH expression. (**C**) Quantification of MAPK-signaling activity. Values were calculated as the ratio between phosphorylated and total ERK signals after normalization to GAPDH levels. Values are the mean ± SEM of four independent experiments. * *p* < 0.05, **** *p* < 0.0001; ns, non-significant (one-way ANOVA followed by Tukey’s multiple comparisons test). (**D**,**E**) RT-qPCR analysis of *SPROUTY2* (**D**) and *SPROUTY4* (**E**) transcript levels in GIST-T1-*Scramble* and GIST-T1-*ShLIX1#2* cells cultured in the absence (-) or presence of 0.5 μM imatinib. Data were normalized to the mean *HMBS* and *YWHAZ* expression. Values are the mean ± SEM of seven independent experiments. * *p* < 0.05, **** *p* < 0.0001; ns, non-significant (one-way ANOVA followed by Tukey’s multiple comparisons test).

**Figure 4 ijms-24-07138-f004:**
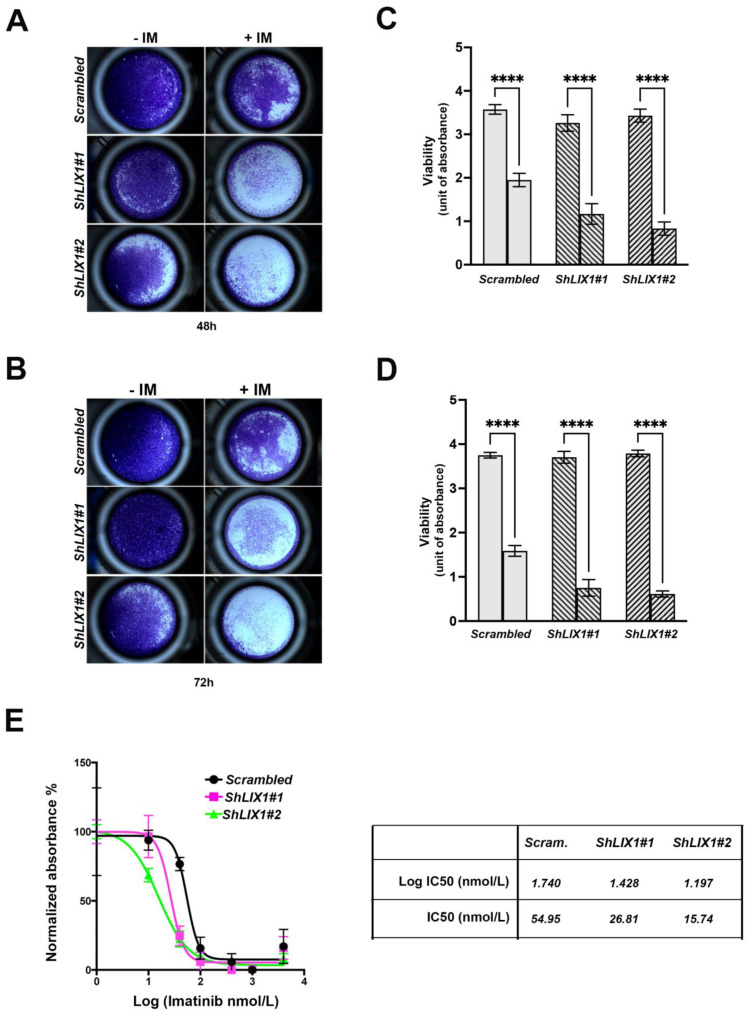
LIX1 silencing enhances imatinib anti-tumor in GIST cells. (**A**,**B**) Crystal violet staining of GIST-T1-*Scramble,* GIST-T1-*ShLIX1#1,* and GIST-T1-*ShLIX1#2* cells incubated (+IM) or not (-IM) with 0.5 μM imatinib. All plates were fixed, stained, and imaged after 48 h (**A**) or 72 h of treatment (**B**). (**C**,**D**) Quantification of cell viability in the different GIST-T1 cell lines at 48 h (**C**) and 72 h (**D**) of treatment. Values are the mean ± SEM of eight independent experiments; **** *p* < 0.0001 (one-way ANOVA). (**F**) Imatinib IC_50_ shows the potentiation of the imatinib effect upon LIX1 silencing in GIST-T1 cells.

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
