# Peer review of "LIX1 Controls MAPK Signaling Reactivation and Contributes to GIST-T1 Cell Resistance to Imatinib"

_ijms, 2023, doi:10.3390/ijms24087138_

Round 1
Reviewer 1 Report
In the paper titled "LIX1 controls MAPK signaling reactivation and contributes to the resistance of gastrointestinal stromal tumors to imatinib" by Salomé Ruiz-Demoulin and colleagues. According to their results, gastrointestinal stromal tumors (GIST), the most common sarcoma, are mostly caused by mutations in the KIT receptor tyrosine kinase. The use of tyrosine kinase inhibitors, such as imatinib and sunitinib, can provide significant benefit; however, in most patients, secondary mutations in KIT will result in failure of the treatment. In order to overcome the emergence of resistance to KIT inhibition, it is imperative that we understand how GIST cells initially adapt to KIT inhibition. A number of mechanisms have been implicated in the resistance to imatinib antitumor effects, including the reactivation of MAPK signaling after KIT/PDGFRA inhibition. It is demonstrated in this study that LImb eXpression 1 (LIX1), a protein we identified as a regulator of the Hippo transducers YAP1 and TAZ, is upregulated in response to the treatment with imatinib or sunitinib. The silencing of LIX1 in GIST cell lines impaired the reactivation of MAPK signaling by imatinib and enhanced its antitumor activity. Regarding the present manuscript, I would like to make a few comments.
The running title should be deleted
The manuscript was enjoyable to read. The in vitro analysis is well described and well conducted. What is the next step in this study?
The results are very encouraging and warrant further investigation. Maybe this is my main criticism of the present study, that it is too brief.
In light of the results of LIX1, what other approximations are possible, since the authors have published other manuscripts on the same gene and evaluated RNA-seq data
Author Response
We would like to thank the reviewer for its constructive and pertinent comments on our manuscript. We have carefully considered its remarks and have modified our manuscript accordingly.
Q: The running title should be deleted
R: The running title has been deleted
Q: What is the next step of the study? In light of the results of LIX1, what other approximations are possible, since the authors have published other manuscripts on the same gene and evaluated RNA-seq data.
R: The experiments presented in this manuscript were performed in a stable GIST cell line in which LIX1 was silenced using the shRNA technology. These results are promising, and we would like now to explore in vivo the effects of inhibiting LIX1 alone or in combination with Imatinib. No model of sporadic GISTs is available so far. Homozygous KIT K641E GIST mice have been developed, but they do not allow investigating the mechanisms of GIST pathogenesis mainly because they phenocopy familial GISTs (1% of all patients with GIST; they develop diffuse ICC hyperplasia that eventually progresses to GIST). Accordingly, these mice display massive ICC hyperplasia in the stomach and GISTs only in the caecum (Sommer et al., 2003). Therefore, we are developing xenografts of Imatinib-sensitive GIST cells in immunodeficient mice to circumvent this limitation (short-time perspective). However, xenograft models using cancer cell lines have several limitations. The first one is that this model may not reflect the patient’s drug response sufficiently. The second issue comes from the observation that this model does not sufficiently represent the complex tumor heterogeneity and the tumor microenvironment. Recent studies demonstrated that in GISTs, the tumor microenvironment, mostly populated by tumor-associated macrophages and lymphocytes, has a significant impact on prognosis and response to treatment (Dimino et al., 2022). For this reason, we will develop PDX models of GIST will be generated by implanting tumor fragments from patients directly into immunodeficient mice (long-time perspective).
Reviewer 2 Report
This study provides evidence that LImb eXpression 1 (LIX1) is upregulated upon imatinib or sunitinib treatment in GIST. This molecule appears to be involved in the tumor resistance to tyrosine kinase inhibitors.
This work is of interest, and it gives some new insight on the response of GIST to therapy. It is well performed and designed.
The main limitation of this manuscript is due to the lack of the analysis of several cell lines. Indeed, the authors performed the experiments on the GIST-T1 and the imatinib-resistant GIST-T1/670.
It is not clear if the reported relevance of LIX is a typical feature of this single cell line (the genetic background is the same) or the reported LIX1 regulation plays a key role in different patients.
The finding that this protein is upregulated in GIST patients is not the evidence that it can display the same behaviour in these patients.
This is a strong limitation of the work, and it should be clearly stated.
The authors have to explain the reason for not having analysed other cell lines and if this manuscript will be accepted, it would be better to indicate the cell line used in the title.
Author Response
We would like to thank the reviewer for its constructive and pertinent comments on our manuscript. We have carefully considered its remarks and have modified our manuscript accordingly.
Q: The main limitation of this manuscript is due to the lack of the analysis of several cell lines. Indeed, the authors performed the experiments on the GIST-T1 and the imatinib-resistant GIST-T1/670. It is not clear if the reported relevance of LIX is a typical feature of this single cell line (the genetic background is the same) or the reported LIX1 regulation plays a key role in different patients. The finding that this protein is upregulated in GIST patients is not the evidence that it can display the same behaviour in these patients. This is a strong limitation of the work, and it should be clearly stated. The authors have to explain the reason for not having analysed other cell lines and if this manuscript will be accepted, it would be better to indicate the cell line used in the title.
R: The primary purpose of this study was to evaluate LIX1 contribution in the early adaptation mechanisms of sensitive-GIST cells in response to Imatinib (First-line treatment). These mechanisms are responsible for the emergence of secondary KIT mutations and resistance. The rebound of MAPK activation is well characterized in Imatinib-sensitive GIST-T1 cells. This cell line is commercially available. The rebound of MAPK activation is also well characterized in Imatinib-sensitive GIST882 cells. This cell line has been developed and characterized by another groups (Tuveson et al. 2001). We agree with the reviewer that it is quite important to investigate LIX1 contribution in MAPK reactivation upon Imatinib treatment in other GIST cell line. The GIST882 cell line is not commercially available and despite our multiple emails, we have not been able to obtain this cell line so far. This obstruction has limited our investigations. In agreement with Reviewer2’s comments, we toned down the interpretation of our results in the conclusion, specifying that this study was carried out only in GIST-T1 cells. We added the following sentence “Here, we provide evidence that LIX1 promotes MAPK reactivation in GIST-T1 cells during treatment with imatinib. Accordingly, LIX1 silencing mimics the effects induced by MAPK inhibitors and enhances imatinib anti-tumor effect. Thus, our work suggests that LIX1 could be a new therapeutic target to prevent MAPK reactivation and overcome TKI resistance in GIST. This research, however, is subject to the main limitation of having evaluated LIX1 only in GIST-T1 cells”.
Reviewer 3 Report
Topic of manuscript is interesting and original with promising applicability in cancer treatment. Nevertheless, some point can be taken for improvement of manuscript. After solving them, I will gladly recommend the acceptance of the manuscript.
Authors should be included figure for the illustration of LIX1 role in imatinib resistance.
Difference between GIST-T1-ShLIX1#1 and GIST-T1-Sh- LIX1#2 should be explained.
Effect on LIX1 silencing was not tested in imatinib resistant cells.
KIT and its role tumorigenesis are mentioned introduction to short.
Role LIX1 in other cancer type should also mentioned,
Author Response
We would like to thank the reviewer for its constructive and pertinent comments on our manuscript. We have carefully considered its remarks and have modified our manuscript accordingly.
Q: Authors should be included figure for the illustration of LIX1 role in imatinib resistance.
R: We added a graphical abstract to illustrate the role of LIX1 in adaptation mechanisms in response to Imatinib
Q: Difference between GIST-T1-ShLIX1#1 and GIST-T1-Sh- LIX1#2 should be explained.
R: The following sentence has been added in the method section. “GIST-T1 cell lines that stably express control shRNA (GIST-T1-Scramble) or shRNAs targeting two distinct regions of LIX1 (GIST-T1-ShLIX1#1 and GIST-T1-ShLIX1#2) were previously developed (Guérin et al., 2020). Their characterization by RT-qPCR analysis have confirmed LIX1 down-regulation in GIST-T1-ShLIX1 cells with a higher efficacy of ShLIX1#2 than ShLIX1#1 (Guérin et al., 2020)”.
Q: Effect on LIX1 silencing was not tested in imatinib resistant cells.
R: The primary purpose of this study was to evaluate LIX1 contribution in the early adaptation mechanisms in sensitive-GIST cells in response to Imatinib (First-line treatment). These mechanisms are responsible for the emergence of secondary KIT mutations and resistance. The rebound of MAPK activation is well characterized in Imatinib-sensitive GIST cells. However, this rebound is poorly studied in Imatinib-resistant GIST cells (such as GIST-T1/670 cell lines) in response to Sunitinib (Second-line treatment). This question is of high importance regarding the emergence of third KIT mutations. It is a novel question that we are investigating now by establishing GIST-T1/670 cell lines that stably express control shRNA (GIST-T1/670-Scramble) and shRNAs against LIX1 (GIST-T1/670-ShLIX1#1 and GIST-T1/670-ShLIX1#2).
Q: KIT and its role tumorigenesis are mentioned introduction to short.
R: The introduction has been re-written accordingly. The following passage has been added. “Oncogenic KIT mutations are found in approximately 80% of sporadic GISTs (Szucs and Jones, 2018), but familial syndromes harboring germline-activating KIT mutations have been described. These patients develop diffuse ICC hyperplasia that eventually progresses to GIST (Miettinen et al., 2006). The KIT proto-oncogene encodes a class III receptor-type tyrosine kinase that is activated upon binding to its cognate ligand, stem cell factor, via its extracellular domain (Klug et al., 2019). This leads to receptor homo-dimerization and activation of the intracellular kinase domain that consequently initializes downstream signaling, such as the PI3K–AKT–mTOR, and RAS–MAPK pathways. These pathways are implicated in regulating cellular functions, especially in ICCs where KIT physiologic activity is indispensable for cell proliferation, differentiation, and apoptosis (Chi et al., 2010). Primary KIT mutations mainly occur in exon 11 (70%–80%) that encodes the juxta-membrane domain. This leads to disruption of the auto-inhibitory function, resulting in constitutive, ligand-independent kinase KIT activity, and constitutive activation of downstream KIT-activated AKT and MAPK signaling. Both pathways are crucial for GIST initiation and tumor development, by exerting a critical regulation of cancer cell proliferation and apoptosis evasion (Bauer et al., 2007; Bosbach et al., 2017; García-Valverde et al., 2020)”.
Round 2
Reviewer 1 Report
Thank you for responding to my previous comments. The main criticism I have is the short presentation of the results in the manuscript. It is important that the authors respond to the various points in a rebuttal letter and do not summarize the ideas in a large paragraph.
Author Response
We would like to thank the reviewer for its comments on our manuscript. We more carefully tried to respond to the various points in this rebuttal letter.
Q: The running title should be deleted
R: The running title has been deleted
Q: What is the next step of the study?
R: The experiments presented in this manuscript were performed in a stable GIST cell line in which LIX1 was silenced using the shRNA technology. These results are promising, and we would like now to explore in vivo the effects of inhibiting LIX1 alone or in combination with Imatinib. No model of sporadic GISTs is available so far. Homozygous KIT K641E GIST mice have been developed, but they do not allow investigating the mechanisms of GIST pathogenesis mainly because they phenocopy familial GISTs (1% of all patients with GIST; they develop diffuse ICC hyperplasia that eventually progresses to GIST). Accordingly, these mice display massive ICC hyperplasia in the stomach and GISTs only in the caecum (Sommer et al., 2003). Therefore, we are developing xenografts of Imatinib-sensitive GIST cells in immunodeficient mice to circumvent this limitation (short-time perspective). However, xenograft models using cancer cell lines have several limitations. The first one is that this model may not reflect the patient’s drug response sufficiently. The second issue comes from the observation that this model does not sufficiently represent the complex tumor heterogeneity and the tumor microenvironment. Recent studies demonstrated that in GISTs, the tumor microenvironment, mostly populated by tumor-associated macrophages and lymphocytes, has a significant impact on prognosis and response to treatment (Dimino et al., 2022). For this reason, we will develop PDX models of GIST will be generated by implanting tumor fragments from patients directly into immunodeficient mice (long-time perspective).
Q: The results are very encouraging and warrant further investigation. Maybe this is my main criticism of the present study, that it is too brief.
R: The primary purpose of this study was to evaluate LIX1 contribution in the early adaptation mechanisms of sensitive-GIST cells in response to Imatinib. In this study, i) we report that LIX1 expression (mRNA and protein) is upregulated in GIST-T1 cells in response of KIT inhibition (induced both by Imatinib and Sunitinib treatment), ii) we reveal that LIX1 expression increase is an early event in GIST cell response to KIT inhibition, we demonstrate that LIX1 iii) is implicated in imatinib-induced MAPK signaling reactivation in GIST-T1 cells and iiii) de-sensitizes GIST-T1 cells to Imatinib. This research, however, is subject to the main limitation of having evaluated LIX1 only in GIST-T1 cells. The rebound of MAPK activation is well characterized in Imatinib-sensitive GIST-T1 cells. This cell line is commercially available. The rebound of MAPK activation is also well characterized in Imatinib-sensitive GIST882 cells. This cell line has been developed and characterized by another groups (Tuveson et al. 2001). It is quite important to investigate LIX1 contribution in MAPK reactivation upon Imatinib treatment in other GIST cell lines. The GIST882 cell line is not commercially available and despite our multiple emails, we have not been able to obtain this cell line so far (while published) probably because of competition issues between research groups (we published on LIX1 and YAP1/TAZ for more than eight years (McKey et al., 2016; Guérin et al., 2020; Guérin et al., 2022)). This obstruction has limited our investigation. We are confident, as the reviewer is, that our results are very encouraging. We are convinced that their publication will help us for further investigations, especially for the in vivo experiments described above.
Q: In light of the results of LIX1, what other approximations are possible, since the authors have published other manuscripts on the same gene and evaluated RNA-seq data.
R: We apologize for not fully understand the question of the reviewer. As mentioned in the manuscript, the main clinical challenges for the forthcoming years is the identification of novel and potent inhibitors that can maximize the response in patients with early-stage disease to avoid the emergence of secondary mutations. Tumor adaptation that mainly involves several interlinked events among which the regulation of mitochondrial metabolism and re-activation of KIT downstream pathways. Blockade of mitochondria activity in addition to KIT (using a mitochondrial OXPHOS inhibitor) i) leads to a switch towards the glycolytic phenotype, ii) impairs MAPK re-activation and iii) re-sensitizes GIST cells to Imatinib (Vitiello et al., 2018). We recently reported that endogenous LIX1 is anchored to the outer mitochondrial membrane in GIST cells. LIX1 is a key regulator of cristae organization, modulating mtROS level and subsequently regulating the signaling cascades that control GIST malignancy. Given the results we have previously published regarding LIX1 role in mitochondrial function (Guérin et al., 2020; Guérin et al., 2022), and because a metabolic shift is suspected to be involved in early adaptation mechanisms (Vietello et al., 2018), we don’t have “other approximations “to do. We have clearly explained in the discussion section that we hypothesize that LIX1 contribution to the adaptation mechanisms of GIST cells to Imatinib relies on its regulative function of mitochondrial metabolism. We expect having responded the question of the reviewer.
Reviewer 2 Report
I think the limitation of having analysed a single cell line is really strong. This should be clearly indicated in the title of the manuscript in order to avoid misinterpretation of the findings reported.
Author Response
We would like to thank the reviewer for its comments on our manuscript.
Q: I think the limitation of having analysed a single cell line is really strong. This should be clearly indicated in the title of the manuscript in order to avoid misinterpretation of the findings reported.
R: It is now clearly indicated in the title of the manuscript that experiments have been done in GIST-T1 cells in order to avoid misinterpretation of the findings reported.
Reviewer 3 Report
I have no objection.
Author Response
We would like to thank the reviewer for its comments on our manuscript.
Round 3
Reviewer 1 Report
I would like to thank the authors for taking into account my comments and clarifying every aspect of the current manuscript. There are no further comments to be made.